# Disentangling the rhythms of human activity in the built environment for airborne transmission risk: An analysis of large-scale mobility data

**Zachary Susswein, Eva C Rest, Shweta Bansal***

Department of Biology, Georgetown University, Washington, DC, United States

## Abstract

**Background:** Since the outset of the COVID-19 pandemic, substantial public attention has focused on the role of seasonality in impacting transmission. Misconceptions have relied on seasonal mediation of respiratory diseases driven solely by environmental variables. However, seasonality is expected to be driven by host social behavior, particularly in highly susceptible populations. A key gap in understanding the role of social behavior in respiratory disease seasonality is our incomplete understanding of the seasonality of indoor human activity.

**Methods:** We leverage a novel data stream on human mobility to characterize activity in indoor versus outdoor environments in the United States. We use an observational mobile app-based location dataset encompassing over 5 million locations nationally. We classify locations as primarily indoor (e.g. stores, offices) or outdoor (e.g. playgrounds, farmers markets), disentangling location-specific visits into indoor and outdoor, to arrive at a fine-scale measure of indoor to outdoor human activity across time and space.

**Results:** We find the proportion of indoor to outdoor activity during a baseline year is seasonal, peaking in winter months. The measure displays a latitudinal gradient with stronger seasonality at northern latitudes and an additional summer peak in southern latitudes. We statistically fit this baseline indoor-outdoor activity measure to inform the incorporation of this complex empirical pattern into infectious disease dynamic models. However, we find that the disruption of the COVID-19 pandemic caused these patterns to shift significantly from baseline and the empirical patterns are necessary to predict spatiotemporal heterogeneity in disease dynamics.

**Conclusions:** Our work empirically characterizes, for the first time, the seasonality of human social behavior at a large scale with a high spatiotemporal resolutio and provides a parsimonious parameterization of seasonal behavior that can be included in infectious disease dynamics models. We provide critical evidence and methods necessary to inform the public health of seasonal and pandemic respiratory pathogens and improve our understanding of the relationship between the physical environment and infection risk in the context of global change.

**Funding:** Research reported in this publication was supported by the National Institute of General Medical Sciences of the National Institutes of Health under award number R01GM123007.

**For correspondence:**
shweta.bansal@georgetown.edu

## Editor's evaluation

This is a valuable study characterizing seasonal deviations in indoor activity at the county level in the United States with relevance to respiratory disease transmission. The strength of evidence is solid. This study and its results are of potential interest to those people constructing more evidence-based infectious disease transmission models.

## Introduction

The seasonality of infectious diseases is a widespread and familiar phenomenon. Although a number of potential mechanisms driving seasonality in directly transmitted infectious diseases have been proposed, the causal process behind seasonality is still largely an open question (*Martinez, 2018*; *Altizer et al., 2006*; *Grassly and Fraser, 2006*). In the case of the influenza virus, seasonal changes in humidity have been identified as a potential mechanism, with drier winter months enhancing transmission (*Shaman and Kohn, 2009*; *Shaman et al., 2010*; *Dalziel et al., 2018*); similar patterns have been observed for respiratory syncytial virus and hand foot and mouth disease (*Baker et al., 2019*; *Onozuka and Hashizume, 2011*). However, humidity is but one of many mechanisms contributing to seasonality in infectious disease transmission. Seasonal changes in temperature, human mixing patterns, and the immune landscape, among other factors, are thought to contribute to transmission dynamics (*Metcalf et al., 2009*; *Mossong et al., 2008*; *Kronfeld-Schor et al., 2021*; *Bakker et al., 2021*; *Altizer et al., 2006*). The relative importance of these disparate mechanisms varies across directly-transmitted pathogens and is still largely unexplained (*Martinez, 2018*; *Grassly and Fraser, 2006*). The influence of seasonal host behavior on respiratory disease seasonality remains particularly understudied (*Fisman, 2012*; *Kronfeld-Schor et al., 2021*) except for a few notable examples (*Bharti et al., 2011*; *Few et al., 2013*; *Kummer et al., 2022*).

For respiratory pathogens spread via the aerosol transmission route, in particular, seasonality may be mediated by multiple behaviorally-driven mechanisms. Aerosol transmission, a significant mode of transmission for a number of respiratory pathogens including tuberculosis, measles, and influenza (*Tellier et al., 2019*), has become increasingly acknowledged during the COVID-19 pandemic (*Greenhalgh et al., 2021*; *Wang et al., 2021*; *Jayaweera et al., 2020*; *Klompas et al., 2020*; *Morawska and Milton, 2020*). The role of aerosols in respiratory disease transmission allows for transmission outside of the traditional 6 ft. radius and 5 min duration for the droplet mode and implicates human mixing in indoor locations with poor ventilation as being a high risk for transmission, regardless of the intensity of the social contact. While more is known about the spatiotemporal variation in environmental factors such as temperature and humidity in the indoor environment (e.g. *Nguyen and Dockery, 2016*) and about the impact these factors have on airborne pathogen transmission (e.g. *Robey and Fierce, 2022*; *Yang and Marr, 2011*), limited information is available on rates of human indoor activity and how this varies geographically and seasonally. In the United States, most studies quantifying indoor and outdoor time are conducted in the context of air pollutants, suffer from small study sizes, lack spatiotemporal resolution, and are outdated. The most cited estimates originate from the 1980 s-90s and estimate that Americans spend upwards of 90% of their time indoors (*Ott, 1988*); more recent data agree with these estimates (*Klepeis et al., 2001*; *Spalt et al., 2016*). While it is well understood that seasonal differences and latitude likely affect time spent indoors, little is known of the spatiotemporal variation in indoor activity beyond this one monolithic estimate, vastly limiting our ability to comprehensively characterize the seasonality of airborne disease exposure risk.

Because our understanding of the drivers of seasonality for respiratory diseases has been limited, the modeling of seasonally-varying infectious disease dynamics has been traditionally done using environmental data-driven or phenomenological approaches. Environmental data-driven approaches incorporate seasonality into epidemiological models through environmental correlates of seasonality, such as solar exposure or outdoor temperature (*Bakker et al., 2021*; *Baker et al., 2019*; *Coletti et al., 2018*). This approach to seasonal dynamics controls for interseasonal variation in transmission dynamics and measures the strength of correlations between proposed metrics and seasonal variation in force of infection – although the observed relationship is rarely causally relevant for respiratory disease transmission. In contrast, phenomenological models such as seasonal forcing approaches modulate transmissibility over time without specifying a particular mechanism for this modulation (*Keeling et al., 2001*; *Altizer et al., 2006*). By applying well-understood functions (such as sine functions), seasonal forcing allows for flexible specification and quantification of dynamics, such as periodicity or oscillation damping, and indirectly captures seasonal variation in nonenvironmental factors such as school mixing. A significant remaining gap in seasonal infectious disease modeling is thus the ability to empirically incorporate spatiotemporal variation in behavioral mechanisms driving seasonality of disease exposure and transmission.

Thus, despite the role of the indoor built environment in exposure to the airborne transmission route, seasonal variation in indoor human mixing has not yet been systematically characterized nor

integrated into mathematical models of seasonal respiratory pathogens. To address this gap, we construct a novel metric quantifying the relative propensity for human mixing to be indoors at a fine spatiotemporal scale across the United States. We derive this metric using anonymized mobile GPS panel data of visits of over 45 million mobile devices to approximately 5 million public locations across the United States. We find a systematic latitudinal gradient, with indoor activity patterns in the northern and southern United States following distinct temporal trends at baseline. However, we find that the COVID-19 pandemic disrupted this structure. Lastly, we fit simple parametric models to incorporate these seasonal activity dynamics into models of infectious disease transmission when indoor activity is expected to be at baseline. Our work provides the evidence and methods necessary to inform the epidemiology of seasonal and pandemic respiratory pathogens and improve our understanding of the relationship between the physical environment and infection risk in light of global change.

## Methods
### Data source
We use the SafeGraph Weekly Patterns data, which provides foot traffic at public locations ('points of interest', hereafter referred to as POIs) across the United States based on the usage of mobile apps with GPS (*Safegraph, 2021a*). The data are from 2018–2020, and 4.6 million POIs are sampled in all years of our study. The data is anonymized by applying noise, omitting data associated with a single mobile device, and is provided at the weekly temporal scale. Data are sampled from over 45 million smartphone devices (of approximately 275–290 million smartphone devices in the United States during 2018–2021 *Statista Digital Market Outlook, 2022*), and does not include devices that are out of service, powered off, or ones that opt out of location services on their devices.

This is secondary data analysis, so no informed consent or consent to publish was necessary. Ethical review for this study (STUDY00003041) was sought from the Institutional Review Board at Georgetown University and was approved on October 14, 2020.

### Defining indoor activity seasonality
Safegraph POIs are locations where consumers can spend money and/or time and include schools, hospitals, parks, grocery stores, restaurants, etc., but do not include home locations. (In *Figure 1—figure supplement 1*, we show that time at home does not display significant seasonal variation). Each POI is assigned a six-digit North American Industry Classification System (NAICS) code in the SafeGraph Core Places dataset to classify each location into a business category. We classify each six-digit NAICS code (363 unique codes in total) as primarily *indoor* (e.g. schools, hospitals, grocery stores) or primarily *outdoor* (e.g. parks, cemeteries, zoos). We classify some locations as *unclear* if the location is a potentially mixed indoor and outdoor setting (e.g. gas stations with convenience stores, automobile dealerships). Approximately 90% of POIs were classified as indoors, 6.5% were classified as outdoors, and 3.5% were classified as unclear. In *Figure 1—figure supplement 2*, we illustrate the robustness of our metric to the classification of unclear locations.

We define $\widetilde{\sigma}_{it}$, *Equation 1*, as the propensity for visits to be to indoor locations relative to outdoor locations. We aggregated raw visit counts, defined when a device is present at a non-home POI for longer than one minute, to all indoor POIs and all outdoor POIs in a given week ($t$) at the US county level ($i$). Visit counts are normalized by the maximum visit counts for indoor or outdoor locations in each county during the year 2019 (In *Figure 1—figure supplement 3*, we show that the maximum visit count is comparable in 2018 and 2019).

$$\widetilde{\sigma}_{it} = \frac{N_{it}^{indoor}/max_t\{N_{it}^{indoor}\}}{N_{it}^{outdoor}/max_t\{N_{it}^{outdoor}\}} \tag{1}$$

This metric is then mean-centered to arrive at a relative measure of indoor activity seasonality, $\sigma_{it}$, which is comparable across all counties:

$$\sigma_{it} = \frac{\widetilde{\sigma}_{it}}{\mu_{\widetilde{\sigma}}} \tag{2}$$

We note that $\mu_{\widetilde{\sigma}}$ is not spatially structured (see *Figure 1—figure supplement 4*).

As a data cleaning step, we use spatial imputation for any county-weeks where sample sizes are small. For location-weeks in which the total visit count is less than 100, we impute the indoor activity seasonality using an average of $\sigma$ in the neighboring locations (where neighbors are defined based on shared county borders). This affects 0.6% of all county-weeks and a total of 79 (out of 3143) counties.

### Time series clustering analysis

To characterize groups of US counties with similar indoor activity dynamics, we use a complex networks-based time series clustering approach. We first calculate the pairwise similarity between z-normalized indoor activity time series for each pair of counties, $i$ and $j$ using the Pearson correlation coefficient ($\rho_{ij}$). For pairs of locations where $\rho_{ij}$ is in the top 10% of all correlations, we represent the pairwise time series similarities as a weighted network where nodes are US counties and edges represent strong time series similarity (In *Figure 2—figure supplement 1*, we show the robustness of our clustering results to this choice of correlation threshold).

We then cluster the time series similarity network using community structure detection. This method effectively clusters nodes (counties) into groups of nodes that are more connected within than between. The resulting clustering thus represents a regionalization of the United States in which regions consist of counties that have more similar indoor activity dynamics to each other than to other regions. One benefit of the network-based community detection approach over other clustering methods is that community detection does not require user specification of the number of clusters (regions, in this case); instead, the number of clusters emerges organically from the data connectivity (*Aggarwal and Reddy, 2013*). For community detection, we use the Louvain method (*Blondel et al., 2008*), a multiscale method in which modularity is first optimized using a greedy local algorithm, on the similarity network with edge weights (i.e. time series correlations) using a igraph implementation in *Python* (*Louvain-igraph, 2018*).

We performed a robustness assessment of the community structure using a set of 25 'bootstrap networks,' $B_i$. For each bootstrap network, the edge weight (i.e. the time series correlation) for each edge of the network was perturbed by $\epsilon\, N(0, 0.05)$. The community structure algorithm was performed on each bootstrap network. A consensus value was then calculated as the sum of the normalized mutual information between the community structure partition of the bootstrap network $B_i$ and all other bootstrap networks. The partition with the largest consensus value was defined as the robust community structure partition.

Given some known limitations to the time series correlation network-based approach to clustering (*Hoffmann et al., 2020*), we validated our network-based clustering results with another common clustering method. In particular, we used hierarchical clustering with Ward linkage and Euclidean distance on z-normalized indoor activity time series, implemented using scipy in *Python*. (We note that Euclidean distance is equivalent to Pearson's correlation on normalized time series *Berthold and Höppner, 2016*). The results of this comparison are summarized in *Figure 2—figure supplement 5*.

### Disruptions to indoor activity due to pandemic response

We investigate the COVID-19 pandemic's impact on indoor activity seasonality by comparing pre-pandemic mobility patterns in 2018 and 2019 with mobility patterns during the COVID-19 pandemic in 2020. We compared the proportion of indoor visits at the county level, $\sigma_{it}$, across 2018, 2019, and 2020 to examine changes in indoor activity seasonality during the COVID-19 pandemic. We also examined total activity, aggregating visits to all indoor, outdoor, and unclear POIs by week and mean-centering them for each US county during the COVID-19 pandemic in 2020.

### Incorporating indoor activity into infectious disease models

We seek to illustrate the impact of incorporating seasonality into an infectious disease model using a phenomenological model versus empirical data. To achieve this, we parameterize a simple compartmental disease model with a seasonality term, using either our empirically-derived indoor activity seasonality metric or an analytical phenomenological model of seasonality fit to this metric.

### Phenomenological model of seasonality

We first fit our empirically-derived indoor activity seasonality metric using a time-varying non-linear model. We specify the time-varying effect as a sinusoidal function as is commonly done to incorporate

seasonality into infectious disease models phenomenologically. The indoor activity seasonality, $\sigma_{it}$ for cluster $i$ at week $t$ is specified as: $\sigma_{it} = 1 + \alpha_i \sin(\omega_i t + \phi_i)$, where $\alpha_i$ is the sine wave amplitude, $\omega_i$ is the frequency, and $\phi_i$ is the phase. We fit a model for locations in the northern cluster separately from those in the southern cluster, as identified above. We fit the parameters for this model using the nlme, a standard package in R for fitting Gaussian nonlinear models.

### Disease model

We model infectious disease dynamics through a simple SIR model of disease spread:

$$\frac{dS}{dt} = -\beta_0 \beta(t) S I$$

$$\frac{dI}{dt} = \beta_0 \beta(t) S I - \gamma I$$

$$\frac{dR}{dt} = \gamma I$$

We incorporate alternative seasonality terms to consider the impact of heterogeneity in indoor seasonality on disease dynamics. For the northern and southern clusters separately, we define modeled seasonality as $\beta(t) = 1 + \alpha \sin(\omega t + \phi)$, with the fitted parameters for each cluster (*Figure 4—figure supplement 1* and *Figure 4—figure supplement 2*). We also consider two exemplar locations for empirical estimates of seasonality, where $\beta(t) = \sigma_t$ after rolling window smoothing: Cook County for an example county from the northern cluster, and Maricopa County for an example location from the southern cluster. We also compare against a null expectation where $\beta(t) = 1$ (All seasonality functions are illustrated in *Figure 4—figure supplement 3*). We assume that $\beta_0 = 0.0025$ and $\gamma = 2$ (on a weekly time scale).

## Results

Based on anonymized location data from mobile devices, we construct a novel metric that measures the relative propensity for human activity to be indoors at a fine geographic (US county) and temporal (weekly) scale. Activity is measured as the number of visits to unique physical, public (non-residential) locations across the United States. Locations are classified as indoors if they are enclosed environments (i.e. buildings and transportation services). We characterize the systematic spatiotemporal structure in this metric of indoor activity seasonality with a time series clustering analysis. We also characterize the shift that occurred in the baseline patterns of indoor activity seasonality during the COVID-19 pandemic. We note that this seasonal variation in the propensity of human activity to be indoors differs from the variation in overall rates of contact or mobility, which does not appear to be highly seasonal (*Figure 1—figure supplement 1*, *Klein et al., 2022*). Lastly, we fit non-linear models to the indoor activity metric at baseline, comparing the ability of a simple model to capture seasonal variation in transmission risk.

### Quantifying empirical dynamics in an indoor activity

The indoor activity seasonality metric, $\sigma$, captures the relative frequency of visits to indoor versus outdoor locations within an area. The components of $\sigma$ capture the degree to which indoor and outdoor locations are occupied; when $\sigma = 1$, a given county is at its county-specific average propensity (over time) for indoor activity relative to outdoor. When $\sigma < 1$, activity within the county is more frequently outdoor and less frequently indoor than average, while $\sigma > 1$ indicates that activity is more frequently indoor and less frequently outdoor than average. Thus, a $\sigma$ of 1.2 indicates that the county's activity is 20% more indoor than average, and a $\sigma$ of 0.80 indicates that the county's activity is 20% less indoor than average (additional details in methods).

Through this metric, we measure the relative propensity for human activity to be indoors for every community (i.e. US county) across time (at a weekly timescale), finding significant heterogeneity between counties (*Figure 1A*). The representative examples of Cook County, Illinois (home of the city of Chicago in the northern US) and Maricopa County, Arizona (home of the city of Phoenix in the southwestern US) highlight systematic spatial and temporal heterogeneity in indoor activity dynamics. In Cook County, indoor activity varies over time, at its peak in the winter, with the relative odds of

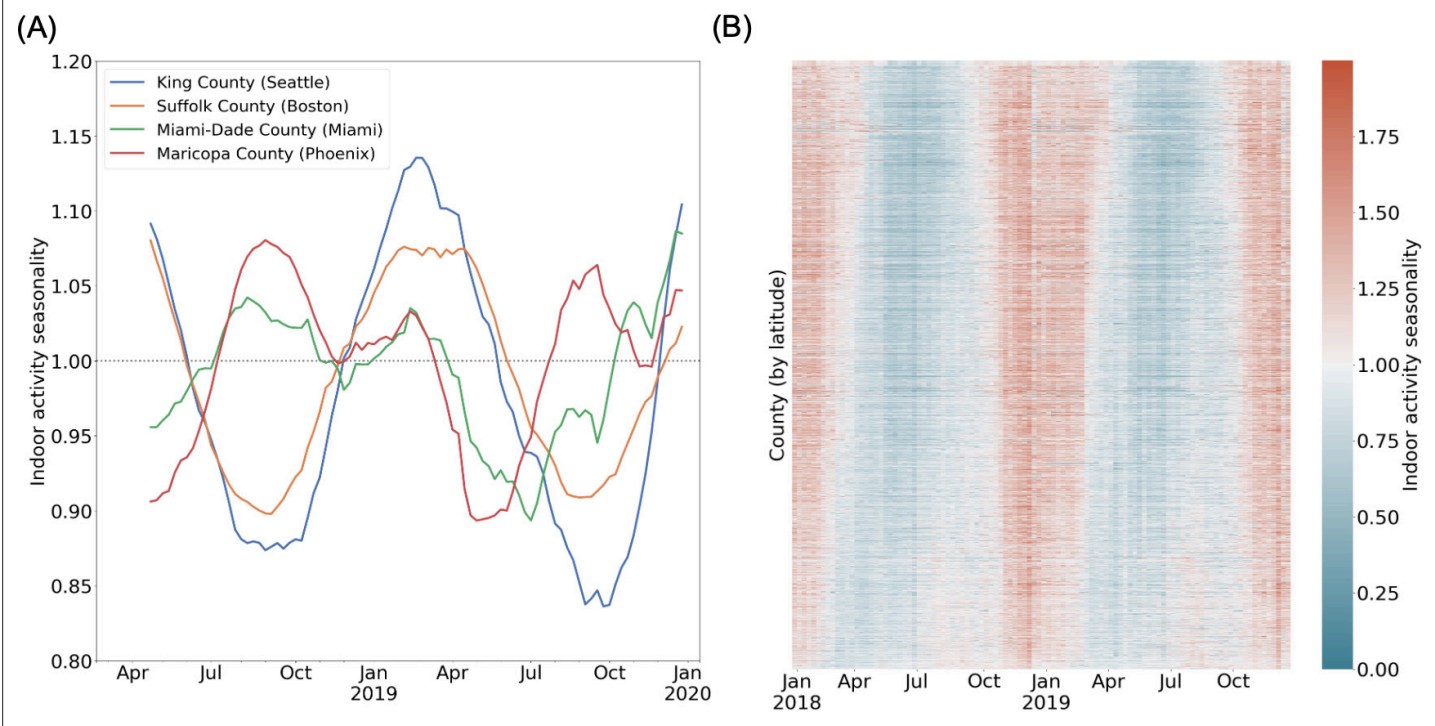

**Figure 1.** Spatio-temporal heterogeneity in indoor activity seasonality. (**A**) Case studies to highlight varying trends in indoor activity seasonality during 2018 and 2019: King County and Suffolk County (in the northern United States) have high indoor activity in the winter months and a trough in indoor activity in the summer months. Miami-Dade and Maricopa County (in the southern United States) see moderate indoor activity in the winter and may have an additional peak in indoor activity during the summer. We apply a rolling window mean for visualization purposes. (**B**) A heatmap of the indoor activity seasonality metric for all US counties by week for 2018 and 2019. Counties are ordered by latitude. We see significant spatiotemporal heterogeneity with distinct trends in the summer versus winter seasons.

The online version of this article includes the following figure supplement(s) for figure 1:

**Figure supplement 1.** Other measures of mobility are not highly seasonal.

**Figure supplement 2.** We demonstrate the effect of the 'unclear' locations on the indoor activity seasonality.

**Figure supplement 3.** We show that the maximum number of visits used in the definition of the $\sigma$ metric is highly comparable in 2018 and 2019.

**Figure supplement 4.** The mean proportion of indoor/outdoor activity ($\mu_{\bar{\sigma}}$) in 2018 displays no latitudinal gradient and is relatively homogeneous across counties; outliers of mean $\geq 2.5$ are removed.

an indoor visit well above average. During the summer, $\sigma$ in Cook County reaches its trough, with activity systematically more outdoors on average. On the other hand, the variation of $\sigma$ across time in Maricopa County is characterized by a smaller winter peak in indoor activity, and an additional peak in the summer (i.e. July and August); this peak occurs concurrently with the trough in Cook County. Unlike in Cook County, $\sigma$ in Maricopa County is lowest in the spring and fall. These representative counties illustrate the systematic within-county variation in indoor activity over time, as well as the between-county variation in temporal trends as represented in *Figure 1B* for all US communities.

To identify systematic geographic structure, we cluster the heterogeneous time series of county-level, weekly indoor activity. We find three geographic clusters corresponding to groups of locations that experience similar indoor activity dynamics (*Figure 2*). These clusters primarily split the country into two clusters: a northern cluster and a southern cluster. Among the communities in the northern cluster, activity is more commonly outdoor over the summer months, trending toward indoor during fall, with a peak in the winter months, as observed in Cook County. Comparatively, the southern cluster has a larger winter peak (i.e. between December and February) and a smaller summer peak (i.e. between July and August); most summer peaks are less extreme than that of Maricopa County (shown). We hypothesize that these two clusters are consistent with climate zones. While there is a moderate association between indoor activity seasonality and environmental variables such as temperature and humidity (*Figure 2—figure supplement 2*), we expect that the northern and southern indoor activity

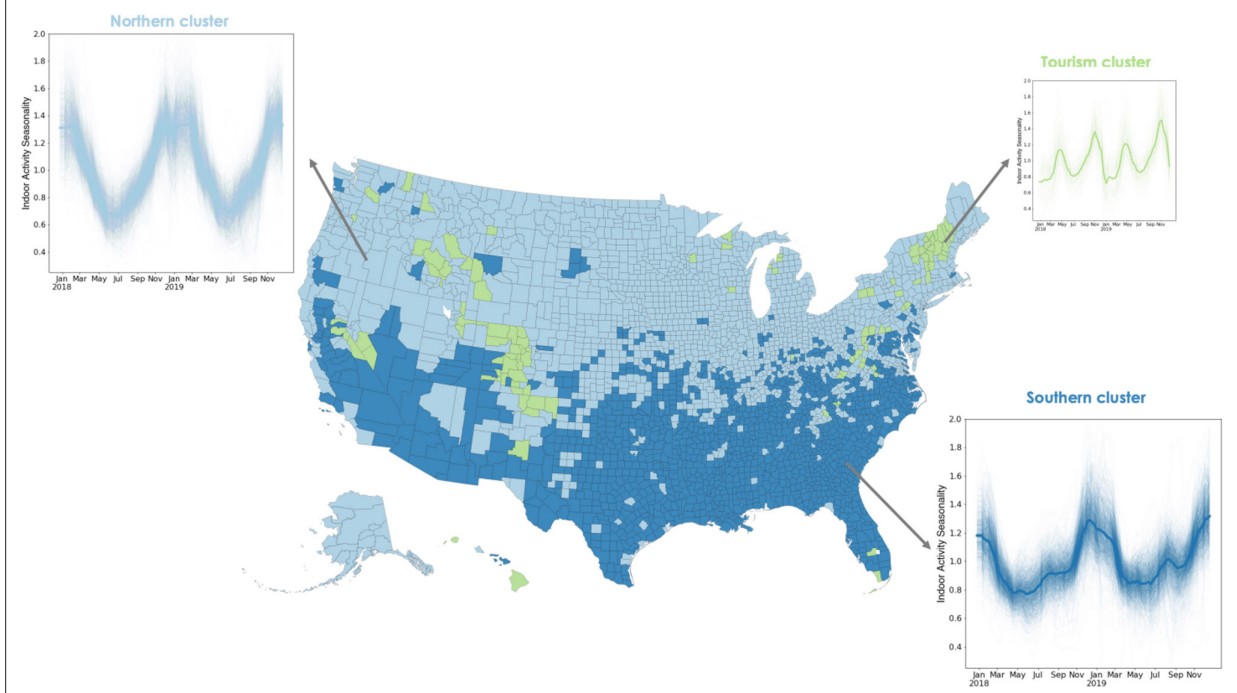

**Figure 2.** Using a time series clustering approach on the indoor activity time series for each US county, we identify groups of counties that experience similar trends in indoor activity. Locations in the northern cluster (light blue) follow a single peak pattern with the highest indoor activity occurring every winter. Locations in the southern cluster (dark blue) experience two peaks in indoor activity each year, one in the winter and a second, smaller one in the summer. The third cluster also experiences two peaks not matching environmental conditions, but potentially corresponding to winter or other tourism areas. We apply a rolling window mean to the time series for visualization purposes.

The online version of this article includes the following figure supplement(s) for figure 2:

**Figure supplement 1.** We illustrate the impact of the correlation threshold on the clustering results (without post-processing).

**Figure supplement 2.** Using data on temperature and rainfall from NOAA's North American Regional Reanalysis (*Mesinger et al., 2006*), we find that indoor activity (sigma) is moderately anticorrelated with both temperature and humidity.

**Figure supplement 3.** Comparison of indoor activity clusters to climate clusters.

**Figure supplement 4.** The third indoor activity cluster displays some correlation with areas of increased tourism, including US ski areas in western and northeastern states, potentially contributing to off-season activity increases.

**Figure supplement 5.** We show the results of time series clustering based on a hierarchical clustering method using Ward linkage and Euclidean distance, implemented using scipy.cluster in *Python*.

clusters will be more consistent with climate zones defined for the construction of the indoor built environment and find that there is indeed substantial consistency between the two (*Figure 2—figure supplement 3*). The third cluster differs substantially: it is geographically discontiguous and its two annual peaks occur during the spring (close to April) and fall (closer to November) seasons. Thus, the counties in this cluster have outdoor activity more frequently than average during both the winter and the summer. The counties in this cluster correspond to locations that are hubs for winter or other tourism, which we speculate is driving their unique dynamics (*Figure 2—figure supplement 4*).

## Characterizing pandemic disruption to baseline indoor activity seasonality

In addition to the description of indoor activity seasonality at baseline, we examine the impact of a large-scale disruption – the COVID-19 pandemic – on these patterns. We compare indoor activity seasonality during the COVID-19 pandemic in 2020 to the baseline patterns of 2018 and 2019. We find that the temporal trends in indoor activity are less geographically structured in 2020 than those of previous years (see *Figure 3—figure supplement 2* for a characterization of the time series patterns). We find that indoor activity deviated from pre-pandemic trends beyond interannual deviations (*Figure 3—figure supplement 1*). We focus on four case studies to highlight the varying impacts

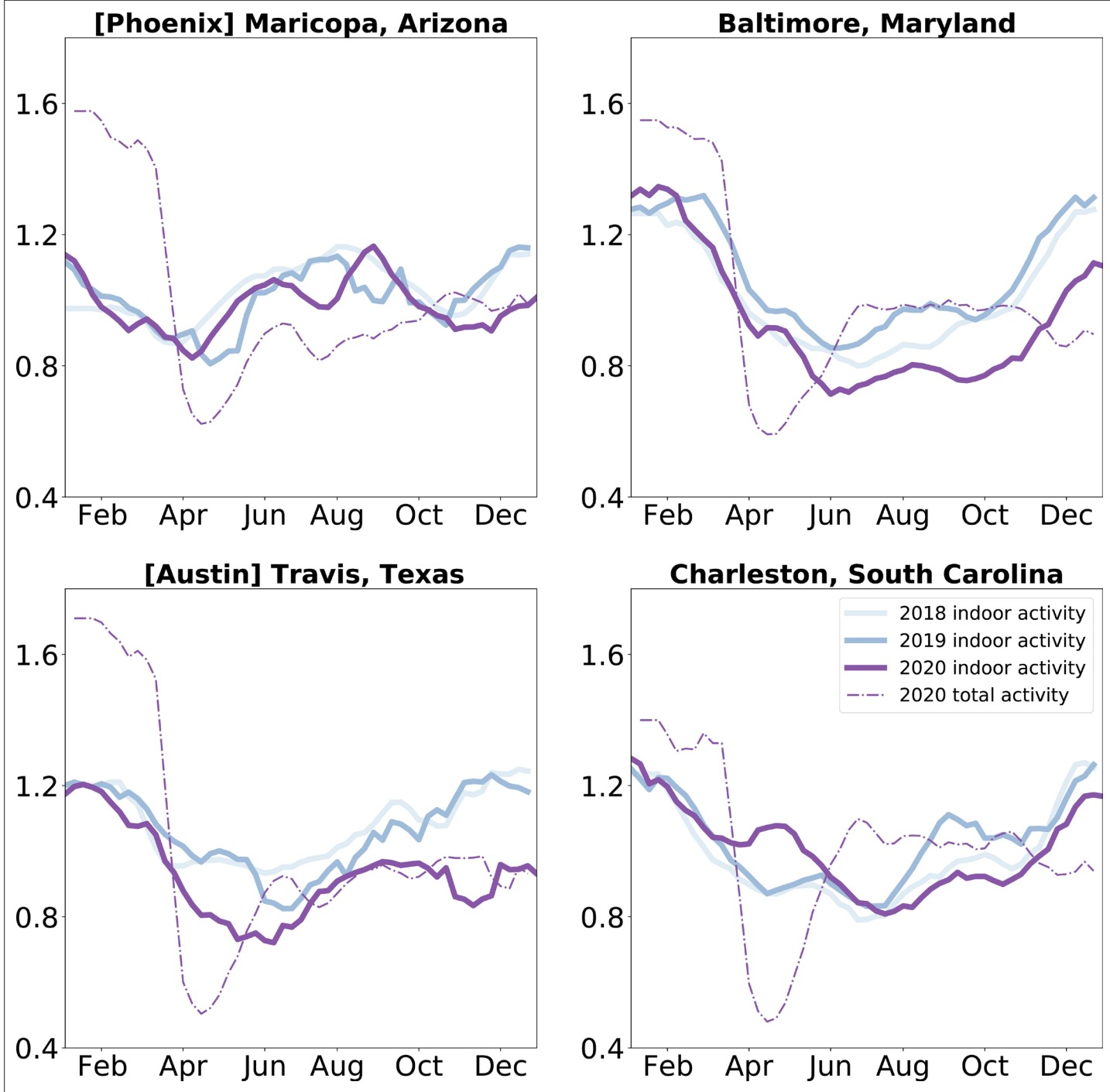

**Figure 3.** Indoor activity during the COVID-19 pandemic was shifted: We compare indoor activity trends in the baseline years of 2018 and 2019 to the pandemic year 2020 in four case study locations. We find that most locations saw a shift in their indoor activity patterns, while others (such as Maricopa County) did not. We also find that while overall activity was diminished uniformly during the Spring of 2020, indoor activity decreased in some locations (Travis County, Texas and Baltimore County, Maryland) and increased in others (Charleston County, South Carolina). We apply a three week rolling window mean to the time series for visualization purposes.

The online version of this article includes the following figure supplement(s) for figure 3:

**Figure supplement 1.** Deviations in 2020 indoor activity from baseline.

**Figure supplement 2.** Indoor seasonality clusters during 2020.

on indoor activity of the pandemic disruption (*Figure 3*). In all four communities, 2020 indoor activity trends shift from 2018 and 2019 patterns, with Maricopa County (home of the city of Phoenix, Arizona) showing the least perturbation relative to prior years. We also find that in early 2020, when there was substantial social distancing in the United States (e.g. school closures, remote work), activity was more likely to be outdoors than in prior years, independent of changes in overall activity levels. With our case studies, we highlight that social distancing policies can have different impacts on airborne exposure risk in different locations: while some locations, such as Travis County (home of Austin, Texas), shifted activities outdoors during this period, reducing their overall risk further, other locations, such as Charleston County (home of Charleston, South Carolina) increased indoor activity above the seasonal average during this period, potentially diminishing the effect of reducing overall mobility. The trends in Charleston are representative of those in the southeastern United States during the spring of 2020 (*Figure 3—figure supplement 1*). By the end of 2020 (and the first winter wave of SARS-CoV-2), many parts of the country were shifting activity more outdoors than seasonally expected (*Figure 3—figure supplement 1*).

## Implications for modeling seasonal disease dynamics

We use this finely-grained spatiotemporal information on indoor activity to incorporate airborne exposure risk seasonality into compartmental models of disease dynamics using common, coarser seasonal forcing approaches. To investigate the impact of heterogeneity in $\sigma$ on the estimation of seasonal forcing for infectious disease models, we fit a sinusoidal model to the time series of indoor activity for each of the primary clusters (*Figure 4A*). We note that because $\sigma$ is defined as deviation from baseline indoor activity, the sinusoidal parameters (amplitude, frequency, phase) should be interpreted as a measure of seasonality in indoor activity, relative to each location's baseline. We find that the parameters of seasonality vary across clusters: the amplitude is higher, and the phase is lower in the northern cluster compared to the southern cluster, indicating a difference in the variability of indoor and outdoor activity seasonality in each cluster (*Figure 4—figure supplement 1*). While the fits are comparable for both clusters (*Figure 4—figure supplement 2*), the sinusoidal model does not capture the second peak of indoor activity during the summer months in the southern cluster. These differences in best fit indicate that sinusoidal models may have an overly restrictive functional form, limiting the accuracy of the approximation, and may underestimate the impacts of seasonality on transmission, obscuring systematic differences between regions. Furthermore, differences in seasonal activity of the observed magnitude can have important implications for disease modeling; applying region-level and county-level forcing to a simple disease model alters incidence patterns (*Figure 4B*). Although region-level seasonality changes incidence timing and peak size relative to a non-seasonal model, it does not fully capture the changes produced by county-level seasonality. These differences indicate that while coarser geographic approximations of seasonality can be appropriate, these approximations can also oversimplify, reducing the accuracy of disease models. Additionally, while simple models of baseline indoor activity can capture seasonality in exposure risk, disruptions such as pandemics can alter this baseline structure and increase heterogeneity.

## Discussion

The seasonality of influenza, SARS-CoV-2, and other respiratory pathogens depends not only on environmental variables but also on the social behavior of hosts. In settings with little prior immunity – such as a pandemic – host social behavior (generating contacts during which transmission may occur) primarily drives heterogeneity in disease dynamic and seasonality is dwarfed by susceptibility (*Baker et al., 2020*). In settings with higher rates of immunity, the contact remains critically important, and seasonal changes in contacts (both direct and indirect) can contribute to the movement of $R_t$ above and below 1 – providing noticeable changes in incidence. Although environmental variables play a role in the seasonality of respiratory pathogens, the role of host social behavior in pathogen seasonality is poorly understood, driven by a poor understanding of indoor versus outdoor social interactions and interactions between behavior and the environment. In this study, we propose a fine-grain measure of indoor activity seasonality across time and space. This metric is a relative quantity of behavior, comparable across locations, and thus intended to be a measure of seasonality beyond a baseline. We determine that indoor activity seasonality displays significant spatiotemporal heterogeneity and that

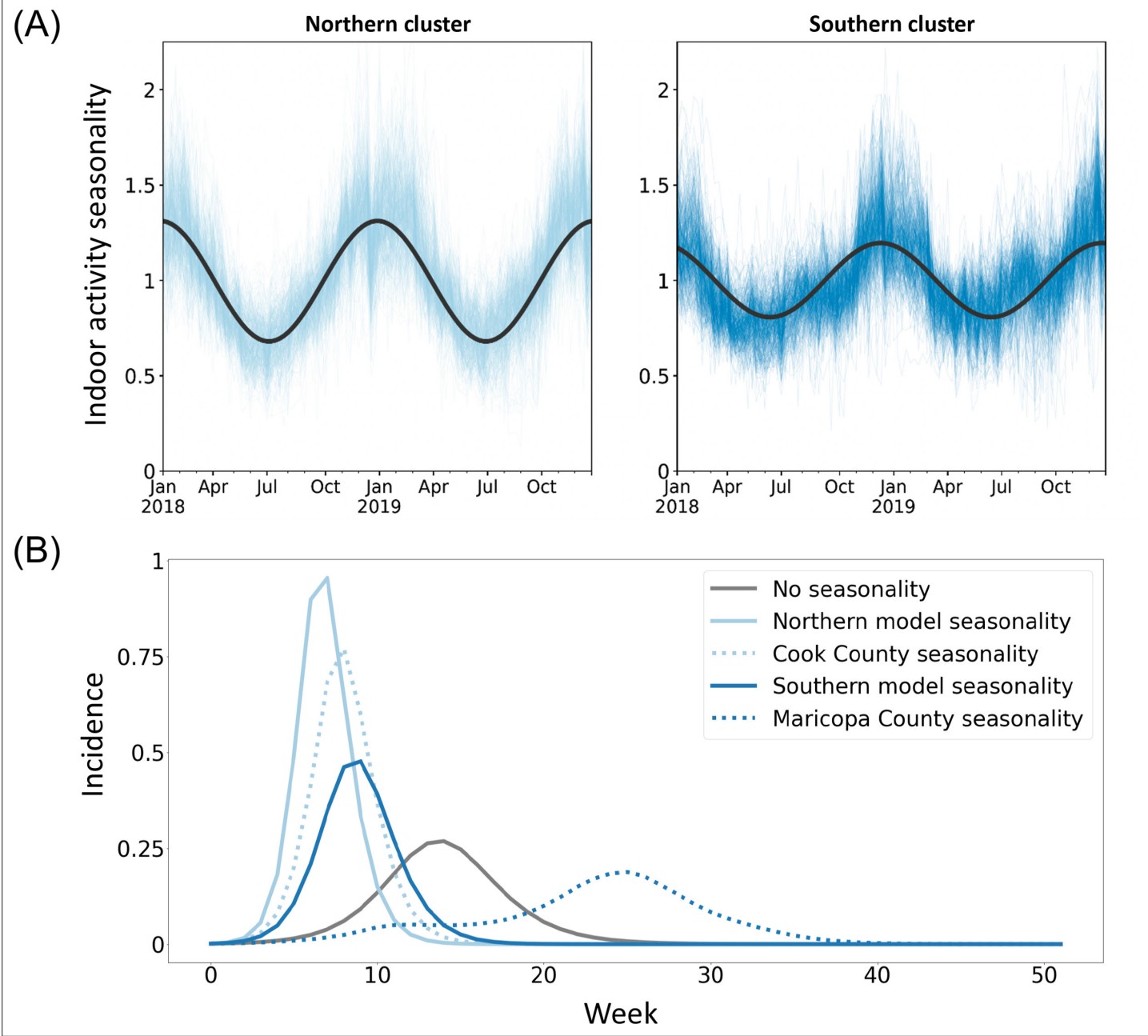

**Figure 4.** Incorporating seasonality in epidemiological models. (**A**) Sine curves fit to the 2018 and 2019 time series data (analogous to seasonal forcing model components) fit the northern cluster better than the southern cluster, with a markedly poorer fit for the southern cluster's second summer peak. (**B**) Regional seasonal forcing models display variation in patterns of disease incidence omitted by a non-seasonal model, but even region-level seasonal forcing does not fully capture within-cluster county-level variation.

The online version of this article includes the following figure supplement(s) for figure 4:

**Figure supplement 1.** Parameters of the sinusoidal model fits.

**Figure supplement 2.** Model performance as measured by the root mean square error of the sine curve fit to the cluster averaged over counties within the cluster.

**Figure supplement 3.** The seasonal forcing functions ($\beta(t)$) we used in the epidemiological model.

this variability is highly geographically structured. We also find that while indoor activity seasonality may be highly predictable under baseline conditions, disruptions such as the COVID-19 pandemic can alter these patterns. Finally, we provide an illustration of how our findings can be incorporated into classical infectious disease models using parsimonious models of exposure seasonality.

The indoor activity seasonality that we quantify may reflect heterogeneity in transmission risk via a number of mechanisms including those affecting host contact, susceptibility, or transmissibility. Increased indoor activity may indicate longer-duration airborne contact (e.g. co-location without direct interaction) between susceptible and infected individuals, elevating respiratory transmission risk. Increased indoor density may also suggest increased droplet contact (e.g. a conversation in close proximity), under homogeneous mixing. Additionally, indoor activity may suggest increased susceptibility as poor ventilation, increased pollutants, reduced solar exposure, and low humidity of the indoor environment have been shown to weaken immune response (*Moriyama et al., 2020*). Finally, increased indoor activity may indicate an increase in transmissibility due to higher exposure as low humidity caused by climate control (heating, ventilation, and cooling, HVAC) in indoor environments has been shown to increase viral survival and HVAC re-circulation has been shown to increase viral dispersion (*Lu et al., 2020*; *Liao et al., 2005*). While our new measure does not disentangle these component mechanisms, it represents an integrated seasonality in exposure risk due to all of these factors and can help lead us to a more complete understanding of the heterogeneity and seasonality in disease dynamics and outcomes.

We find that spatiotemporal heterogeneity in the indoor activity metric can be decomposed into two large geographically-contiguous groups in the northern and southern United States representing distinct temporal dynamics in indoor activity. These groups closely correspond to built environment climate zones, potentially explaining this systematic variability. We note, however, that while these clusters overlap with climate classifications, this correspondence does not suggest that environmental variables such as temperature and humidity should be used to represent behavioral heterogeneity. Climatic factors within these climate zones may be related to, but not necessarily correlated with, the seasonality of human mixing within these zones. Additionally, even in the case that environmental factor variability drives behavioral variability, it would be critical to capture the effect of behavior on disease directly so as to not obscure any direct effects of climatic factors on disease.

We illustrate how to incorporate seasonality in exposure risk to future models of disease dynamics using a simple phenomenological model. We use this traditional model of infectious disease dynamics to evaluate the implications of the spatial coarseness of seasonal forcing. Our results suggest that the substantial local heterogeneity in the dynamics of indoor activity across time and space could be large enough to alter seasonality in infectious disease dynamics. While our work does not consider observed transmission patterns, we suggest that researchers carefully consider the spatial scale on which they model seasonality in theoretical models, commonly used for scenario analysis and model-based intervention design (e.g. *Borchering et al., 2021*). We additionally highlight that the use of simple or complex functional forms of seasonality requires statistical fits to baseline data and, in the case of disruptions, these fitted models may no longer be appropriate. . Although indoor activity is moderately anticorrelated with temperature and humidity (*Figure 1*), weather-derived covariates are not able to completely reflect the impacts of human movement (but they may have some statistical power). We show patterns of human mobility changed substantially during the COVID-19 pandemic, potentially contributing to changes in infectious disease seasonality.

Recent work during the COVID-19 pandemic demonstrates the impact of reduced occupancy in indoor locations and increasing outdoor activity on the likelihood of disease transmission. In particular, behavioral interventions or nudges that reduce occupancy are more impactful than reducing overall mobility as they reduce visitor density and the likelihood of density-dependent airborne tnsmission (*Chang et al., 2021*). Similarly, the availability of outdoor areas in urban settings, such as public parks, has been demonstrated to reduce case rates when population mobility becomes less restricted (*Johnson et al., 2021*). Our results suggest that such public health strategies should be implemented in a targeted manner, informed by real-time data, and with clear communication of the goals. We found notable changes occurred in indoor activity seasonality at the start of the COVID-19 pandemic, despite relatively consistent patterns during the spring season in prior years. Designing a behavioral strategy and measuring its effectiveness without real-time data could thus be misleading. Our finding of two distinct geographic clusters of indoor activity suggests the need for geographical targeting of

strategies to reduce indoor transmission risk. While northern latitudes might benefit from decreased indoor occupancy and increased outdoor activity in Northern Hemisphere winters, southern latitudes should be additionally targeted for such interventions in the summer months. Lastly, our findings highlight the need to communicate the goals of behavioral interventions clearly. While all communities universally reduced overall activity during the early days of the COVID-19 pandemic, some increased indoor activity during this time, potentially diminishing the positive effects of the social distancing policies put into place. A public health education campaign to clarify the role of indoor interactions in transmission risk may have ameliorated this.

Our study leverages a novel data stream made available to researchers due to the COVID-19 pandemic. Similar datasets are available globally, part of a $12 billion location intelligence industry (*Keegan and Ng, 2021*). Such novel data streams offer many opportunities to address long-unanswered questions in infectious disease and climate change behavior dynamics, but these data must be interpreted carefully. Safegraph's mobile-app-based location data does not include data on individuals less than 16 years of age (*Safegraph, 2021b*). While we may expect that children under 12 may be accompanied by adults that may be represented in the dataset, our metric likely does not capture the activity dynamics of older children (children 12–15 make up 5% of the US population). For those included in the Safegraph database, representation is dependent on smartphone usage and a number of business processes not transparent to users of the data, thus we expect that there is geographic variation in the representativeness of the data. Smartphone ownership has increased in recent years, with 85% of US adults reporting smartphone ownership; however, smartphone usage does vary significantly by age, with only 61% of adults over 65 reporting smartphone use (*Pew Resesarch Center, 2021*). Additionally, data shows that location sharing among mobile users is not significantly biased by age, gender, race/ethnicity, income, or education (with 40–65% of all demographic groups participating in location sharing) (*Zickuhr and Smith, 2011*). Based on an analysis done by Safegraph, the panel is representative of race, educational attainment, and income (*Fox, 2019*). On the other hand, a recent independent analysis shows that older and non-white individuals are less likely to be captured in the panel for POI-specific analyses (*Coston et al., 2021*). It is important to note that both studies are associative in nature as the devices in the panel are fully anonymized, so no device-level demographic data exists. Continued work to understand the sampling biases of such datasets will be needed so that improved bias correction approaches can be developed (*Coston et al., 2021*). Additionally, we limit our scope in this study to consider only the number of visits and do not incorporate information about visit duration. The dataset counts all visits of 1 minute or longer. For disease transmission, there may be a threshold duration required for an interaction between an infected and susceptible individual for infection to be propagated. These thresholds are not well-understood for all respiratory diseases, but evidence that SARS-CoV-2 transmission can occur with brief encounters has emerged (*Pringle et al., 2020*). While the Safegraph dataset does provide median dwell times for POIs, the likely significant heterogeneity in the distribution of dwell times remains unknown and is difficult to capture in an aggregated manner.

Our metric and analysis also focus on the US county scale to reflect the finest scale generally used for infectious disease modeling as well as public health decision-making. This choice is likely to ignore some within-county heterogeneity and means that our metric does not represent the experience of all groups, particularly by socioeconomic status. For example, low-income and racially marginalized communities have systematically less access to outdoor, natural spaces and spend more time indoors due to structural inequities including lack of paid leave (*Spalt et al., 2016*; *Nesbitt et al., 2019*; *Sefcik et al., 2019*). Such socioeconomic disparities have been further exacerbated during the pandemic, which potentially affects our indoor activity estimates during 2021. Thus, our estimate of a county's indoor transmission risk may represent an underestimate of the risk experienced by individuals in these communities. We commit to continued work to better characterize the transmission risk experienced by vulnerable populations. Lastly, we acknowledge that data modeling work that can influence public health policy decisions, particularly during an ongoing crisis, must be done with care to prevent misconceptions from having adverse effects on risk perception and policies (*Carlson et al., 2020*). We thus strongly note that while our measure of indoor behavioral seasonality provides a potential driver of respiratory disease seasonality, it remains one among many complex factors which integrate to predict the transmission potential of an ongoing epidemic or pandemic (*Susswein et al., 2021*). Thus, we cannot rely on behavioral seasonality to diminish transmission naturally, and

pandemic intervention strategies should not be planned around behavioral seasonality while population susceptibility remains high in so many locations.

Ongoing global change events highlight the importance of this work, as it informs how widespread disruptions may shift patterns of indoor activity, potentially altering traditional infectious disease seasonality. Climate change events will continue to cause significant disruption to normal behavior patterns; mechanistic understanding of infectious disease seasonality and real-time data collection will be crucial components of future disease control efforts. While other global change events may impact indoor activity in different ways than the COVID-19 pandemic, a rigorous understanding of the impact of host behavior on infectious disease allows policymakers and emergency preparedness experts to effectively address future disruptions.

## Acknowledgements

Research reported in this publication was supported by the National Institute of General Medical Sciences of the National Institutes of Health under award number R01GM123007. The content is solely the responsibility of the authors and does not necessarily represent the official views of the National Institutes of Health. We gratefully acknowledge the data sharing by Safegraph which made this study possible. We thank Alexes Merritt for her data processing efforts.

## Additional information

### Competing interests

Zachary Susswein: is currently employed at the Rockefeller Foundation as a Data Analyst. The author has no other competing interests to declare. The other authors declare that no competing interests exist.

### Funding

| Funder | Grant reference number | Author |
| --- | --- | --- |
| National Institute of General Medical Sciences | R01GM123007 | Zachary Susswein Eva C Rest Shweta Bansal |

The funders had no role in study design, data collection and interpretation, or the decision to submit the work for publication.

### Author contributions

Zachary Susswein, Software, Formal analysis, Validation, Investigation, Visualization, Methodology, Writing – original draft, Writing – review and editing; Eva C Rest, Data curation, Validation, Investigation, Writing – review and editing; Shweta Bansal, Conceptualization, Resources, Data curation, Software, Formal analysis, Supervision, Funding acquisition, Validation, Investigation, Visualization, Methodology, Writing – original draft, Project administration, Writing – review and editing

### Author ORCIDs

Zachary Susswein http://orcid.org/0000-0002-4329-4833
Eva C Rest http://orcid.org/0000-0002-6461-3450
Shweta Bansal http://orcid.org/0000-0002-1740-5421

### Ethics

Human subjects: Ethical review for this study was sought from the Institutional Review Board at Georgetown University and the study was approved on October 14, 2020 (STUDY00003041). This is secondary data analysis, so no informed consent or consent to publish was necessary.

### Decision letter and Author response

Decision letter https://doi.org/10.7554/eLife.80466.sa1
Author response https://doi.org/10.7554/eLife.80466.sa2

## Additional files

### Supplementary files
• MDAR checklist

### Data availability
We make available on Github the data and code needed to reproduce all figures and analyses in this manuscript: https://github.com/bansallab/indoor_outdoor (copy archived at swh:1:rev:d8a2ffc-49f46a22c45814bd1dfcd1b054f2a4a27). The dataset we provide is of the metric used in all our analyses and figures ("indoor activity"). This dataset can be regenerated using the Safegraph Weekly Patterns datasets found at https://docs.safegraph.com/docs/weekly-patterns and code in the Github repository. The Safegraph Weekly Patterns was made freely available to academics at a uniquely granular level in response to the COVID-19 pandemic. Safegraph's business model involves selling these datasets to other corporations and, as a result, any data access agreement with the company forbids sharing of the raw data. The company does, however, make its data freely available to academics (for non-commercial use) through an institutional university subscription to Dewey or an individual data use agreement with Safegraph.

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
