## [Editor Report]

This is a valuable study characterizing seasonal deviations in indoor activity at the county level in the United States with relevance to respiratory disease transmission. The strength of evidence is solid. This study and its results are of potential interest to those people constructing more evidence-based infectious disease transmission models.

---

## [Decision Letter]

**Decision letter after peer review:**

Thank you for submitting your article "Disentangling the rhythms of human activity in the built environment for airborne transmission risk: a large-scale analysis of mobility data" for consideration by *eLife*. Your article has been reviewed by 3 peer reviewers, and the evaluation has been overseen by a Reviewing Editor and Diane Harper as the Senior Editor. The following individual involved in the review of your submission has agreed to reveal their identity: Guillaume Béraud (Reviewer #3).

Essential revisions:

1) One of the major issues is related to the definition of "indoor". It should be stated from the beginning what "indoor" locations are about, in particular, that home is not part of it, which is an important point to understand the results (and the discussion). Did the authors conduct any sensitivity analysis related to these choices?

2) Compare trends with other proxies for seasonality.

3) Provide a better justification of the clustering approach used (choice of methodology) and conduct the necessary sensitivity analyses (including choice of threshold).

4) Compare model fits to real COVID-19 data and/or ILI (because of the difficulty in dealing with NPIs for COVID-19), or step back from the claims that the 'σ' metric generates better model fits given that you've shown that it yields different model fits, but not necessarily better ones.

5) Provide formal justifications for several claims throughout the Results section (e.g. a measurement of geographic heterogeneity in trends, and differences in pre-pandemic vs. peri-pandemic patterns).

In summary, this work has potential but requires essential revisions, particularly by clarifying some of the language used and by including more detailed and quantitative analyses.

*Reviewer #1 (Recommendations for the authors):*

Overall, I found this study to be well-conducted and impactful. There are a number of areas where it could be improved, particularly by clarifying some of the language and by including more detailed and quantitative analyses. In the following comments, I'll try to provide specific suggestions for where the authors might focus their efforts.

Specific comments:

10: Perhaps "impacting" rather than "suppressing"? The rest of the manuscript makes clear that the authors interpret seasonality as a force that can both enhance and suppress transmission, so it would be worth using consistent language here.

54-56 ("While more is known about… rates of indoor activity.") I had trouble parsing this sentence. What do you mean by spatio-temporal variation in the indoor environment? At what scale? Are you talking about the environment itself, or the variation in people's experience of it? What kinds of rates of indoor activity are unknown, and why does this matter? It seems to me that this sentence is identifying the key gap that this study aims to fill, so it would be worth making this more precise.

90: What sort of global change do you mean? As it stands, the term is too vague to be meaningful here.

Figure 1: The authors could consider grouping Figure 1B by latitude rather than alphabetically; this might reveal some interesting patterns that more clearly support their findings of different seasonal patterns in the north vs. the south.

109: systematic how? This paragraph and Figure 1A seem to discuss just two counties, but a "systematic" difference suggests to me that there's some kind of variation that's observed repeatedly across instances (counties). It would be good to discuss here what exactly is systematic about this variation.

Figure 2: Did the data include Alaska and Hawaii and/or the territories? I would imagine these states might also have substantially different seasonal trends relative to the lower 48 and might give some indication of what sorts of seasonal trends we might expect outside the US.

140: Heterogeneous how? Can you provide some measurement of the degree of heterogeneity?

142: ("in most locations indoor activity deviated from pre-pandemic trends") – again, as measured how? In what fraction of locations? To what degree did the trends deviate?

146: ("activity was more likely to be outdoor than in prior years") – how much more likely? How widespread was this change?

158-159: It would be worth reporting the changes in amplitude and phase here, with appropriate units.

161: A poorer fit as measured how?

170: "accuracy of disease models" – I appreciate that the authors have shown that using different seasonal forcing terms as inputs can yield different epidemic curves, but I don't think they've made a formal assessment of accuracy here, which would require comparing model fits to disease transmission data. The evidence presented so far does not make clear to me the degree of detail needed in the seasonal forcing term to accurately characterize disease transmission trends, as this will depend on the disease dynamics themselves and the temporal and geographic scale of the model.

234: Perhaps "southern latitudes should be targeted for such interventions in summer months as well"? It seems that southern latitudes still have a substantial winter peak, it's just that they also have a summer peak.

278: Again, it would be worth specifying what global change events the authors have in mind here.

Another general point for the Discussion: how should we interpret differences in amplitude across locations? Since σ is a measurement of the percent change in baseline activity, the indoor activity in a location with a high baseline but low σ might still be higher than the indoor activity in a location with a low baseline but a large σ. To what extent can we use σ to compare indoor activity across locations in the US? Or can we only use it to compare variations in indoor activity within counties? Would it be worth including some analysis of the baseline indoor activity across the US, since σ is really operating on this baseline?

288: Make explicit that you'll be referring to this as a POI.

294: What kind of spatial imputation did you do? Why?

Figures S7: It feels odd to me to have amplitude, frequency, and phase all plotted on the same vertical axis despite them all having different units. Perhaps a table would be better?

*Reviewer #2 (Recommendations for the authors):*

I am confident that a revision of the issues in the public review would improve the quality of the paper and allow it to exploit the full potential of this work.

I believe that it is crucial to repeat the analysis taking into account the nature of the correlation matrix, so either adjusting the null hypothesis on modularity optimization or by using a different community detection algorithm.

*Reviewer #3 (Recommendations for the authors):*

Overall, it is an excellent paper and very well written. However, there are some issues I'd like to be considered:

The major issue is related to the definition of "indoor". It should be stated from the beginning what is "indoor" locations, in particular, that home is not part of it, which is an important point to understand the results (and the discussion). At the moment, it is only defined within the methods (Line 300), at the end of the article.

Line 43-41: Maybe authors could extend a few references on seasonality causes. But it is not mandatory.

Figure 1B: Why counties are ordered in alphabetical order? Which does not bring a lot of information, while it could be ordered by latitude, as an example, which could reveal some patterns.

Lines 103-106: authors should define more precisely what is the average (county-level? season? …).

Cluster: Maybe the clustering methods could be explained more extensively in the appendix.

Figure 3: I found it difficult to observe a shift in indoor activities, according to season.

Line 232-233: Isn't it the contrary? Increase of indoor activities in winter for northern regions?

Finally, a discussion on the difference between relationship and causality could be useful to distinguish human behavior seasonality and infectious diseases seasonality.

---

## [Author Response]

Essential revisions:1) One of the major issues is related to the definition of "indoor". It should be stated from the beginning what "indoor" locations are about, in particular, that home is not part of it, which is an important point to understand the results (and the discussion). Did the authors conduct any sensitivity analysis related to these choices?

We have now added text to the Results to define indoor, and added more details to the Methods section to provide further clarity on our classification of locations. Additionally,

– We have now added a sensitivity analysis to the Supplement (Figure S11) that shows that the locations that could not be classified as indoor or outdoor do not make a significant impact on the measure of indoor activity seasonality.

– We have added a figure (Figure S1) that shows that time spent at home does not appear to be highly seasonal.

2) Compare trends with other proxies for seasonality.

We have now added a supplementary figure (Figure S3) to compare trends in indoor activity seasonality with those in environmental variables such as temperature and humidity.

3) Provide a better justification of the clustering approach used (choice of methodology) and conduct the necessary sensitivity analyses (including choice of threshold).

We have now added significant details to the Methods section on the clustering approach, and added sensitivity analysis and methodological comparison to the Supplement (Figures S13, S14).

4) Compare model fits to real COVID-19 data and/or ILI (because of the difficulty in dealing with NPIs for COVID-19), or step back from the claims that the 'σ' metric generates better model fits given that you've shown that it yields different model fits, but not necessarily better ones.

We appreciate this point, and agree that future work will have to consider how indoor activity seasonality affects our ability to capture observed transmission trends. However, such work would additionally need careful characterization of other seasonal factors hypothesized to drive transmission (including environmental and other behavioral factors), and is beyond the scope of our work. Instead, in Figure 4, we aim to (a) provide the infectious disease modeling community with empirically-inferred parameters for a simple sinusoidal model which is commonly used in infectious disease models to capture transmission seasonality; and (b) demonstrate the implications of ignoring geographic heterogeneity in transmission seasonality in theoretical models of disease dynamics, which are commonly used for scenario analysis and model-based intervention design. As we demonstrate, transmission seasonality described by such sinusoidal models, even when they are empirically characterized as in our case, can lead to meaningfully different epidemic dynamics when transmission seasonality varies from the assumptions. We have added text in the Results and Discussion sections to clarify these points.

5) Provide formal justifications for several claims throughout the Results section (e.g. a measurement of geographic heterogeneity in trends, and differences in pre-pandemic vs. peri-pandemic patterns).

We have added additional clarifying text or statistical analyses to respond to these reviewer points.

Reviewer #1 (Recommendations for the authors):Overall, I found this study to be well-conducted and impactful. There are a number of areas where it could be improved, particularly by clarifying some of the language and by including more detailed and quantitative analyses. In the following comments, I'll try to provide specific suggestions for where the authors might focus their efforts.

We thank the reviewer for their positive feedback on our work.

Regarding the geographic scope and generalizability of findings:

– We acknowledge that our study is limited to the US because of the data available to us. However, similar data can be obtained by researchers globally and our metric and its insights will be generalizable to these datasets. We have now added a sentence (with a reference) to our Discussion section.

– We acknowledge that analysis is limited to those mobile users that share location data. We already include in our Discussion section an acknowledgement that smartphone usage varies by age. Additionally, research shows that location sharing among mobile users is not significantly biased by age, gender, race/ethnicity, income or education within the United States (with 40-65% of all demographic groups participating in location sharing). We have added this information and a reference to our Discussion.

While it’s important to acknowledge these biases, we believe that the benefits of leveraging these data to provide insights into built environment social behavior dynamics outweigh these limitations.

Specific comments:10: Perhaps "impacting" rather than "suppressing"? The rest of the manuscript makes clear that the authors interpret seasonality as a force that can both enhance and suppress transmission, so it would be worth using consistent language here.

We have edited the Background section of the Abstract based on this suggestion.

54-56 ("While more is known about… rates of indoor activity.") I had trouble parsing this sentence. What do you mean by spatio-temporal variation in the indoor environment? At what scale? Are you talking about the environment itself, or the variation in people's experience of it? What kinds of rates of indoor activity are unknown, and why does this matter? It seems to me that this sentence is identifying the key gap that this study aims to fill, so it would be worth making this more precise.

Thank you for this suggestion. We have edited this sentence in the Introduction.

90: What sort of global change do you mean? As it stands, the term is too vague to be meaningful here.

Here, we refer to the field of global change which focuses on the planetary-scale biological changes occurring due to anthropogenic impacts.

Figure 1: The authors could consider grouping Figure 1B by latitude rather than alphabetically; this might reveal some interesting patterns that more clearly support their findings of different seasonal patterns in the north vs. the south.

We have re-designed Figure 1 to order the heatmap in panel B according to latitude. Additionally, we have added more data to panel A to provide more intuition.

109: systematic how? This paragraph and Figure 1A seem to discuss just two counties, but a "systematic" difference suggests to me that there's some kind of variation that's observed repeatedly across instances (counties). It would be good to discuss here what exactly is systematic about this variation.

Thank you for this suggestion. We have edited this sentence in the Results to not include the word “systematic”.

Figure 2: Did the data include Alaska and Hawaii and/or the territories? I would imagine these states might also have substantially different seasonal trends relative to the lower 48 and might give some indication of what sorts of seasonal trends we might expect outside the US.

We had previously limited our analysis to the continental US, but have now added Alaska and Hawaii to the results in Figures 1 and 2, as well as the supplementary figure maps. Alaska clusters with the Northern Cluster in Figure 2, while Hawaii’s islands demonstrate more dynamic indoor activity patterns with some islands grouping into the tourism cluster.

140: Heterogeneous how? Can you provide some measurement of the degree of heterogeneity?

Thank you for this point. We have edited the sentence to be more precise:

“We find that the temporal trends in indoor activity are less geographically structured in 2020 than those of previous years (see Supplementary Figure S7 for a characterization of the time series patterns).”

142: ("in most locations indoor activity deviated from pre-pandemic trends") – again, as measured how? In what fraction of locations? To what degree did the trends deviate?146: ("activity was more likely to be outdoor than in prior years") – how much more likely? How widespread was this change?

These are excellent points. We have edited this section of the manuscript for clarity. Additionally, we now (a) report statistics to quantity the deviation in the 2020 indoor activity in the manuscript, pointing to new supplementary table S1; (b) quantify the locations in which activity shifted indoors vs outdoors during two period of the COVID-19 pandemic in 2020 (new Figure S6).

158-159: It would be worth reporting the changes in amplitude and phase here, with appropriate units.

We have updated Supplementary Figure S9 (previously Figure S7) to which we have added differences between clusters in the inferred parameters and included the units.

161: A poorer fit as measured how?

We thank the reviewer for this helpful question. We have added a figure to the supplement (Figure S10) that shows model performance (in terms of root mean square error). We have also added this statement to the relevant Results section:

“While the fits are comparable for both clusters (Supplementary Figure S10B), the sinusoidal model does not capture the second peak of indoor activity during the summer months in the southern cluster.”

170: "accuracy of disease models" – I appreciate that the authors have shown that using different seasonal forcing terms as inputs can yield different epidemic curves, but I don't think they've made a formal assessment of accuracy here, which would require comparing model fits to disease transmission data. The evidence presented so far does not make clear to me the degree of detail needed in the seasonal forcing term to accurately characterize disease transmission trends, as this will depend on the disease dynamics themselves and the temporal and geographic scale of the model.

We appreciate this point by the reviewer, and agree that future work will have to consider how indoor activity seasonality affects our ability to capture observed transmission trends. However, such work would additionally need careful characterization of other seasonal factors hypothesized to drive transmission (including environmental and other behavioral factors), and is beyond the scope of our work. Instead, in Figure 4 we aim to (a) provide the infectious disease modeling community with empirically-inferred parameters for a simple sinusoidal model which is commonly used in infectious disease models to capture transmission seasonality; and (b) demonstrate the implications of ignoring geographic heterogeneity in transmission seasonality in theoretical models of disease dynamics, which are commonly used for scenario analysis and model-based intervention design. As we demonstrate, transmission seasonality described by such sinusoidal models, even when they are empirically characterized as in our case, can lead to meaningfully different epidemic dynamics when transmission seasonality varies from the assumptions.

We have added text to our Discussion paragraph (starting with “We illustrate how to incorporate seasonality…”) to clarify these points.

234: Perhaps "southern latitudes should be targeted for such interventions in summer months as well"? It seems that southern latitudes still have a substantial winter peak, it's just that they also have a summer peak.

Thank you for this point. We agree, and have edited the sentence to clarify this:

“…, southern latitudes should be additionally targeted for such interventions in the summer months.”

278: Again, it would be worth specifying what global change events the authors have in mind here.

Please see our previous response. We have edited some of the mentions of “global change” to “climate change” in this paragraph for additional clarity.

Another general point for the Discussion: how should we interpret differences in amplitude across locations? Since σ is a measurement of the percent change in baseline activity, the indoor activity in a location with a high baseline but low σ might still be higher than the indoor activity in a location with a low baseline but a large σ. To what extent can we use σ to compare indoor activity across locations in the US? Or can we only use it to compare variations in indoor activity within counties? Would it be worth including some analysis of the baseline indoor activity across the US, since σ is really operating on this baseline?

Yes this is correct. Our definition of indoor activity seasonality (σ) is intended to be a *relative* measure. Our focus is on *seasonality*, which measures deviations from a baseline, even if the baseline may be different for different locations. We have now added a sentence to the Discussion to highlight this point (fourth paragraph) and to the Results subsection titled “Implications for modeling seasonal disease dynamics”.

288: Make explicit that you'll be referring to this as a POI.

Thanks. We’ve edited the sentence to clarify this.

294: What kind of spatial imputation did you do? Why?

We’ve added the following sentence to the methods:

“For location-weeks in which the total visit count is less than 100, we impute the indoor activity seasonality using an average of σ in the neighboring locations (where neighbors are defined based on shared county borders). This affects 0.6% of all county-weeks and a total of 79 (out of 3143) counties.”

Figures S7: It feels odd to me to have amplitude, frequency, and phase all plotted on the same vertical axis despite them all having different units. Perhaps a table would be better?

We have now added a table to the updated figure (Figure S9), and added the units and quantified the differences in the caption of this figure.

Reviewer #3 (Recommendations for the authors):Overall, it is an excellent paper and very well written. However, there are some issues I'd like to be considered:The major issue is related to the definition of "indoor". It should be stated from the beginning what is "indoor" locations, in particular, that home is not part of it, which is an important point to understand the results (and the discussion). At the moment, it is only defined within the methods (Line 300), at the end of the article.

Thank you for this point. We have now added text to the Results to define indoor, and added more details to the Methods section. Additionally, we have now added a sensitivity analysis to the Supplement (Figure S11) that shows that the locations that could not be classified as indoor or outdoor do not make a significant impact on the measure of indoor activity seasonality. Also, we have added a figure (Figure S1) that shows that time spent at home does not appear to be highly seasonal.

Line 43-41: Maybe authors could extend a few references on seasonality causes. But it is not mandatory.

Our original list of references covers a diverse set of factors including environmental and social that are discussed in the literature, thus we have not added additional references.

Figure 1B: Why counties are ordered in alphabetical order? Which does not bring a lot of information, while it could be ordered by latitude, as an example, which could reveal some patterns.

We have re-designed Figure 1 to order the heatmap in panel B according to latitude. Additionally, we have added more data to panel A to provide more intuition.

Lines 103-106: authors should define more precisely what is the average (county-level? season? …).

We have added a clarification to that sentence to specify that it’s a county-specific average over time.

Cluster: Maybe the clustering methods could be explained more extensively in the appendix.

We have added more details on the clustering methods in the Methods and Supplement (Figures S13). Additionally, we have now added comparison to other clustering methods to demonstrate the robustness of our results (Supplementary Figure S14).

Figure 3: I found it difficult to observe a shift in indoor activities, according to season.

We have now provided in the Supplement (Table S1 and Figure S6) additional summary statistics to characterize the deviation in indoor activity seasonality during 2020 from earlier years.

Line 232-233: Isn't it the contrary? Increase of indoor activities in winter for northern regions?

We’re not sure we follow this point, so please feel free to elaborate. This sentence is about the design of public health intervention strategies in which transmission risk may be lowered by decreasing indoor activity and/or increasing outdoor activity during winters in northern areas (where indoor activity is naturally high during this period).

Finally, a discussion on the difference between relationship and causality could be useful to distinguish human behavior seasonality and infectious diseases seasonality.

We agree with the reviewer on this important point. We have made an effort to make this point in our Discussion section in a few ways:

– In the second paragraph, we discuss the many underlying mechanisms via which indoor activity seasonality may reflect seasonality of infectious disease. Some of these mechanisms are directly causal (e.g. high indoor activity leads to increased contact which leads to increased transmission of airborne diseases), while others are indirectly causal or simply correlative (e.g. indoor activity suggests increased transmissibility due to poor ventilation).

– In the third paragraph of the Discussion section, we make a case for considering behavior seasonality and environmental seasonality in disease models as each factor has the potential to affect disease dynamics independently or via an interaction.